# Effects of Artificially Induced Breast Augmentation on the Electromyographic Activity of Neck and Trunk Muscles during Common Daily Movements

**DOI:** 10.3390/jfmk7040080

**Published:** 2022-10-03

**Authors:** Christina Kateina, Dimitris Mandalidis

**Affiliations:** Sports Physical Therapy Laboratory, Department of Physical Education and Sports Science, School of Physical Education and Sports Science, National and Kapodistrian University of Athens, 17237 Athens, Greece

**Keywords:** breast volume, external breast implants, postural control, EMG, erector spinae, upper trapezius, everyday movements

## Abstract

A female breast can be a potential source of musculoskeletal problems, especially if it is disproportionately large. The purpose of the present study was to examine the effect of artificially induced breast volume on the EMG activity of neck and trunk musculature during common everyday movements. The EMG activity of the sternocleidomastoid (SCM), the upper trapezius (UT), and the thoracic and lumbar erector spinae (TES, LES) were recorded during 45° trunk inclination from the upright standing and sitting postures (TIST45°, TISI45°) as well as during stand-to-sit and sit-to-stand (STSI, SIST) in 24 healthy females with minimal and ideal breast volume (M-NBV, I-NBV). All movements were performed before and after increasing M-NBV and I-NBV by 1.5-, 3.0-, 4.5-, and 6-times using silicone-gel implants. Significantly higher EMG activity for TES and LES were found at 6.0- and ≥4.5-times increase the I-NBV, respectively, compared to smaller breast volumes during TIST45°. EMG activity of UT was higher, and TES was lower in M-NBV females compared to I-NBV females in all movements but were significantly different only during SIST. The female breast can affect the activity of neck and trunk muscles only when its volume increases above a certain limit, potentially contributing to muscle dysfunction.

## 1. Introduction

Breast volume is constantly subjected to small fluctuations during a females’ adult life. Once the female’s breasts reach their final volume, further hormonal-induced changes that occur during the menstrual cycle [1], pregnancy and lactation [2] as well as menopause [3], may temporarily increase their volume. Physically, the breasts can also become enlarged from the adverse side effects of combined oral contraceptive pills [4] or in response to body weight fluctuations [5]. The volume of a female’s breast may eventually be excessive, as in cases of macromastia a condition that is characterized by pathologically enlarged breast volumes [6]. Artificially, on the other hand, the female’s breast can be subjected to sudden and often extreme enlargement as in surgically implanted silicone gel- or saline-filled breast implants for improving or restoring the aesthetics of their body. In these cases, a female’s breast volume can be increased with breast implants, the volumes of which can range from 175–225 mL [7]; that is, volumes corresponding to an increase of the bra size by 1-cup, up to about 600 mL [8], depending on a female’s body type and the preferences of clients/patients. However, although breast volume is of most concern to females as it relates to their image and aesthetics, it is the weight of the breast that can affect their skeletal and soft tissue structures, especially in females with large breasts, causing musculoskeletal pain syndromes and dysfunction. Considering that the breast’s mass-to-volume ratio is close to one for both premenopausal (1.07 g/mL) and postmenopausal breasts (1.06 g/mL) [9], it becomes apparent that the load added by the breasts on the female’s chest may vary from a few grams to several kilograms depending on their volume. In cases of the congenital absence of breasts, breast removal or breasts with minimal volume, the potential loads applied on the female’s chest can be negligible. Considering that for every kilogram of weight gained the weight of each breast increases by 20 g, these loads can be increased by approximately 150 g if the body weight increases by 7–8 kg [5]. Similar is the weight that is artificially added after breast augmentation using a 150 mL silicone gel- or a saline-filled breast implant considering that their mass-to-volume ratio is 0.97 g/mL and 1.00 g/mL, respectively [10]. However, the added weight on the chest may reach up to 3.0-kg in females with macromastia (2 breasts × 1500 mL volume each) [6] or with artificially enlarged breasts, if a pair of the largest silicone gel implants approved by the FDA (800 mL) for use in breast augmentation surgery are selected [11].

Several studies so far have shown that continuous application of breast mass, especially when it is great, can significantly affect the females’ musculoskeletal system, causing over time adjustments in posture, such as forward trunk tilt [12], kyphosis [13,14], and weakness in the muscles of the upper torso [15]. These muscles are supposed to counteract the mass of physical oversized or excessive prosthetic breasts in a manner similar to that of trying to compensate for the mass of an object carried in front of the trunk [16,17,18]. Clinically, these conditions can be manifested with neck [19], thoracic [15] and back pain [20] possible due to overload and fatigue of the trunk extensors.

However, the research on the effects of different breast volumes on a female’s body is limited [21], with most studies focusing on healthy females with large breasts or patients who have undergone breast reduction surgery. Recent findings, on the other hand, have shown that based on bra size, the females’ breast has been increased in the general population over the past two decades, from 34B to 34DD [22]. Weight gain and breast augmentation surgical procedures that have also increased by 48% between 2000 and 2018 [23] have played a significant role in this increase. These changes can affect not only the musculoskeletal system but also the females’ general health, as the multi-planar breast movements that occur during different types of exercise [24] may prevent them from participation in physical activities [25]. Avoiding such activities may lead to a further increase of body weight and eventually the mass of the breasts. Knowing the potential effects of breast volume/mass on the musculoskeletal system may enable clinicians to design rehabilitation programs to prevent the onset of clinical symptoms or to treat individuals with apparent musculoskeletal problems. The purpose of the present study therefore was to examine the effect of artificially induced breast augmentation on the EMG activity of the neck and trunk musculature during common everyday movements.

## 2. Materials and Methods

### 2.1. Sample

Twenty-four healthy female college students 19–35 years of age with minimal (M-NBV, *n* = 13) and ideal (I-NBV, *n* = 11) as well as symmetrical breast volumes (<50 mL difference between breasts) participated in the study (Table 1). The sample size was determined based on an a priori power analysis aiming to achieve a statistical power of 80%, with an effect size (f) = 0.25 (calculated based on a partial *η^2^* = 0.06) and a statistical significance of a = 0.05, by implementing a free accessed statistical application (G*Power v. 3.1.9.2; FranzFaul, Universität Kiel, Kiel, Germany, accessed on 5 January 2022). The M-NBV group included females whose volume of each breast ranged between 50 mL and 150 mL. It was assumed that the breast volume in this group approximated the small breast volume experienced by females with micromastia or after mastectomy. Females with I-NBV were considered to be those with a breast volume between 200 mL and 250 mL. The breast volume of these subjects was within the limits of what is considered ideal (200–350 mL) based on clinical observations of the volumes used to fill the implants in breast reconstruction surgery [6,26]. Individuals with (i) musculoskeletal asymmetries (e.g., trunk scoliosis with ≥5.0° trunk rotation during execution of the Adam’s test or leg length discrepancy ≥0.5 cm) and/or (ii) musculoskeletal disorders (e.g., previous, or present pathology/injury in the trunk as well as at the upper or lower limbs) were excluded from the study. The study protocol was approved by the Human Research Ethics Committee of the Department of Physical Education and Sports Science of the National and Kapodistrian University of Athens’s (Reg. No. 1335/15-12-2021), and each participant signed an informed written consent prior to testing.

### 2.2. Testing Procedure

All participants visited the research facility twice. The first time each volunteer provided information regarding medical history and undergone breast volume measurements as well as a thorough musculoskeletal evaluation. Only individuals who met the criteria for participation in the experimental procedure underwent surface electromyography in a second visit. Following instructions regarding the experimental process, each participant performed in a random order four guided movements: (i) trunk inclination from the upright standing posture at 45° with knees straight (TIST45°), (ii) trunk inclination from the upright sitting posture at 45° (TISI45°), (iii) stand-to-sit (STSI), and (iv) sit-to-stand (SIST) (Table 2). The intended angle of trunk inclination (45°), either from the upright standing or the sitting posture, was determined using an inclinometer. The required angle of trunk inclination was achieved by allowing contact of the upper limit of the breasts with a telescopic antenna that was extended at an appropriate distance across the chest of each participant. The antenna was connected to a stadiometer that enabled height adjustments according to each participant’s anthropometry. For the movements performed in the sitting posture, the height of the seat was adjusted to achieve 90° of knee flexion. These movements were chosen because they are very common in everyday life and expected to challenge adequately trunk musculature [27]. Each movement was divided into individual phases (starting and final position as well as transition from the starting to the final and from the final to the starting position), with each one lasting three seconds. All movements were performed without and with increasing the female M-NBV and I-NBV by 1.5, 3.0, 4.5, and 6.0-times, at a rate of 60 bits/s using a metronome and repeated three times in each one of the testing conditions. The order in which the movements were performed as well as the order in which the breast volume increased to perform these movements was randomly determined by generating two sets of random sequences using an online application (https://www.random.org accessed on 20 December 2021).

#### 2.2.1. Breast Volumetric Calculations

Breast volume was calculated anthropometrically based on the formula V = 1/3 × 3.14 × MP^2^ × (MR + LR + IR − MP), where MP the mammary projection—that is, the horizontal distance between the thoracic wall and the nipple—and MR, LR and IR the distances be-tween the nipple and the medial terminal crest (medial breast radius), the lateral terminal crest (lateral breast radius) and the inframammary fold (inferior breast radius), respectively [28]. Breasts’ distances were measured with a standard tape measure in centimeters. The mammary projection was calculated by subtracting the horizontal distance measured from a fixed point to the nipple from the horizontal distance measured from the same fixed point to the thoracic wall using a laser distance meter (PLR 50, Robert Bosch GmbH, Leinfelden–Echterdingen, Germany). The high reliability and validity that this procedure has demonstrated was considered appropriate for clinical use [29].

Breast volume was increased using five pairs of silicone-gel-based, teardrop-shape external implants with various factory-determined volumes (Mentor Corp., Santa Barbara, CL, USA). This type of breast implant (anatomical) was used as the geometrical differences that they present in comparison to the round shape implants (e.g., lower point of maximum projection compared to the round shaped implants) provide aesthetically, according to several surgeons, more natural results. The masses of the implants as they were measured with a digital scale were almost equal to their volumes giving eventually a mass-to-volume ratio almost equal to one. The projection of each implant was measured with a caliper (Figure 1).

After calculating the volume that would increase an individual’s breasts to a specific volume, the appropriate combination of implants was placed over each breast where they were secured with a special elastic, non-adjustable in terms of straps tension, bra worn by each volunteer (Table 3). Bags filled with water, the volume of which was measured with a syringe, were used when the volume of the available implants exceeded or was insufficient to increase the desired breast volume. In fact, these custom-made water-filled “implants” were designed in such a way (more flat than bulky) that they do not increase the actual volume of the breasts and therefore their projection, but only the mass of the breast based on their volume (considering that 1 g = 1 mL of water). The projection of these “implants” therefore was negligible and for the purpose of this study considered equal to zero.

#### 2.2.2. EMG Recordings

The EMG activity of sternocleidomastoid (SCM), the upper trapezius (UT) and the thoracic and lumbar erector spinae (TES and LES) was recorded unilaterally (on the side of the dominant upper limb), using an MP 100 Biopac System recording device (BiopacMP 100, Aero Camino Goleta, CA, USA). Pairs of disposable self-adhesive disc-shape (0.9-cm in diameter) Ag-AgCl electrodes (Red Dot™ type 2223, 3M Health Care, St Paul, MN, USA) were placed 2 cm apart in the direction of muscle fibers of (i) the SCM muscle belly, 2 cm from the muscle insertion onto the mastoid process [30] (ii) the UT muscle in the midsection of the line joining the C7 spinous process and the acromion [31], (iii) the TES at a distance of 4 cm from the thoracic process of the T9 vertebra [32] and (iv) the LES on the straight line joining the L1-L2 intervertebral space with the caudal tip of the posterior superior iliac spine at the level of L4-L5 interspace [33]. A ground electrode was placed on each participant’s clavicle. Prior to electrode placement, the skin surface was abraded with ethylic alcohol to reduce skin impedance. The signal was amplified using a differential amplifier and data was recorded at a sampling rate of 2000 Hz. The raw EMG signals were band-pass-filtered (FIR) between 10–500 Hz. A High Pass (FIR) filter (50 Hz) was implemented to reduce the heartbeat’s related noise. A digital camera (LifeCam VX 2000, 1.3 MP, 30-Hz, Microsoft Corporation, Redmond, WA, USA) was synchronized with the EMG recording device to monitor and visualize the movements as they executed. Data acquisition and analysis was performed using the hardware-related computer software (AcqKnowledge, v. 3.9.1.6, Biopac Systems, Inc. Aero Camino Goleta, CA, USA).

#### 2.2.3. Signal Processing and Data Analysis

The raw EMG signals that were recorded in each repetition and testing condition, as identified on the recorded video, were processed into root mean square (RMS) data using a window of 30-ms. Data was analyzed based on the average EMG activity that was recorded during the middle two seconds in each one of the three repetitions for each testing condition. The mean EMG activity of the three repetitions was included in the analysis.

### 2.3. Statistical Analysis

All the data were checked for normality with the Shapiro–Wilk test followed by visual inspection of the Q-Q plot and the box plot. Violations of statistical assumptions for normality necessitated a logarithmic transformation of EMG signals to improve normality and to pull univariate outliers close to the center of distribution for data analysis purposes. Logarithmic means and standard deviations were back transformed and presented as geometric means and a 95% confidence interval. The homogeneity of variance for the between-participants variable was examined with the Levene’s Test. Sphericity was determined based on the Mauchly’s Test, and the Greenhouse–Geisser correction was used when sphericity was significant.

A mixed model ANOVA with interactions was implemented to assess the differences between M-NBV and I-NBV females (between-subjects) and breast augmentations within each group of females (within-subjects) of the mean EMG activity elicited during each movement investigated. Significant main effects were followed by pairwise comparisons after controlling for type I errors using a Bonferroni adjustment. Statistical analyses were conducted in SPSS Statistics for Windows, Version 28.0. (IBM Corp., Armonk, NY, USA), and the level of significance was set at *p* ≤ 0.05.

## 3. Results

Repeated measures of ANOVA revealed significant size main effects (F = 8.322, *p* ≤ 0.001, *η^2^* = 0.274) and marginally significant group-by-breast volume interaction (F = 2.457, *p* = 0.051, *η^2^* = 0.100) for the EMG activity of the TES during TIST45°. Significant were also the breast volume main effects (F = 5.692, *p* ≤ 0.001, *η^2^* = 0.206) and the group-by-breast volume interaction (F = 2.962, *p* < 0.05, *η^2^* = 0.119) for the EMG activity of LES during the same movement.

A post hoc analysis revealed significantly higher EMG activity for TES in females with 6.0-times increase the I-NBV and for LES in females with 4.5- and 6.0-times increase the I-NBV compared to females with smaller breast volumes in the same group, (see Figure 2 for pairwise comparisons). The differences between breast volumes in females with M-NBV regarding the EMG activity of TES and LES, and between breast volumes in both groups regarding the EMG activity of SCM and UT, were not significant.

The EMG activity of TES and LES was greater, and that of UT was lower in females with I-NBV compared to females with M-NBV, with no apparent differences between the two groups regarding the EMG activity of SCM during TISI45°. However, both the within and between group differences were not significant (Figure 3).

No significant within and between group differences regarding breast volume were also found for all the muscles tested during STSI (Figure 4). In contrast, significant group main effects were found for the EMG activity of TES during SIST, being significantly higher in females with I-NBV compared to females with M-NBV (F = 5.424, *p* < 0.05, *η^2^* = 0.198). Significant breast volume main effects were found for the EMG activity of UT (F = 9.158, *p* < 0.001, *η^2^* = 0.294). Females with I-NBV generally showed lower UT activity compared to females with M-NBV, except when breast volume increased 6.0-fold where muscle activity was higher compared to smaller breast volumes in the same group, and similar with those of the group of women with M-NBV (see Figure 5 for pairwise comparisons).

## 4. Discussion

Results of the present study revealed that neck and trunk extensors respond differently to breast volume increase in females with M-NBV and I-NBV while they were performing movements that are part of daily activities. Increasing breast volume in females with M-NBV did not affect muscle activity in any of the movements investigated as opposed to females with I-NBV who demonstrated increased activation of certain muscles with increased breast volume at least in some of the movements tested. More specifically, activation of trunk extensors, i.e., TES and LES, was greater when breast volumes increased 4.5- and 6.0-times in females with I-NBV during TIST45° compared to females in the same group with smaller breast volumes. The contribution of TES and LES in counteracting the increased bending moment produced by the increased breast mass and resistance lever arm length generated by the anterior displacement of the upper body’s center of mass [18] can be justified by the architectural and biomechanical characteristics of the thoracic and lumber erector spinae as well as the multifidus muscle. These muscles, although their level arm is reduced during trunk inclination [34], they can effectively resist the bending forces and the associated shearing and compressive forces generated in the lumbar spine both via storing elastic energy as muscle length increases with lumbar spine flexion [35], and optimization of their length-tension relationship, that is increased linearly up to 45° trunk forward bending [36].

Eventually, the mass of the implants required to elicit compensatory responses of trunk extensors was approximately 0.80 kg, and 1.15 kg; that is, the additional masses used to increase the I-NBV by 4.5- and 6.0-times, respectively. These masses corresponded to 1.4% and 2.0% of the participants body weight and were much smaller than those implemented in previous studies, where similar responses were elicited with loads of various types and shapes (e.g., school bag, barbell, or a barrel) but with masses weighing between 6.5–14.0 kg representing 10–20% of participants’ body weight [16,17,18]. In one of these studies the authors suggested that the female’s breasts may account for the greater EMG activity of the trunk extensors presented by females when carried an external load in front of their chest compared to males [18]. To the best of the authors’ knowledge, this is one of the few, if not the only, study to present evidence showing the effect of such low masses as those corresponding to female breast in the activity of trunk extensor muscles.

The absence of any additional dynamic response by the antigravity posterior muscles with smaller breast volumes (females with ≤3-times increase the I-NBV and with M-NBV), suggest that the moments produced by the masses that corresponded to these breast volumes may have been compensated by the passive stabilization structures of the posterior trunk. The global lumbar passive tissue moment produced by these passive structures reaches approximately 100 Nm as the trunk leans freely forward to 45° from the upright standing posture, regardless of speed (3, 5, or 7 s) with which the movement is executed [37], and this is substantiated by the passive forces generated at different angles of lumbar flexion. For instance, the ligamentum flavum would be expected to generate tension throughout the range of lumbar flexion given its resting length is occurred near the upright standing posture [38]. Tension is also expected to begin to be generated by the supraspinous and interspinous ligaments near the mid-range of trunk flexion [38], and by the articular ligament at about 4 degrees of L4/L5 flexion angle [39].

The passive moment produced by the corresponded structures of the posterior trunk may also prevent trunk extensors to increase their EMG activity as breast volume increased in females with M-NBV and I-NBV during both TISI45°, STSI and SIST. Yet, the activation patterns between groups were different, albeit these differences were not always significant, with the activity of the thoracic trunk extensors being higher and the activity of the upper trapezius being lower in females with I-NBV compared to their counterparts with M-NBV. These responses were elicited having established a relatively high activity of the trunk extensors by instructing participants to sit upright and try to keep their trunks as straight as possible facing forward thus maintain a certain amount of head protraction throughout execution of the movements under investigation. Performing TISI45°, STSI and SIST this way was deemed necessary to achieve a uniform execution of the movements between the participants since different sitting postures and head protraction postures may affect trunk extensors [40] and upper trapezius neck activity [41]. The higher activity of the thoracic trunk extensors generated during these movements and more importantly during SIST in females with INMV was probably attributed to the greater contraction required by these muscles to counteract the bending moment produced by the trunk inclination and the greater mass of the larger breast implants applied on the chest compared to the females with M-NBV. The fact that larger breast implants projected forward more than smaller breasts (up to 4.5-times compared to 3.6-times the natural breast projection) may have ultimately brought the center of mass of the upper body closer to the base of support, thus requiring less trunk flexion for transition to the standing posture compared to females with M-NBV [42]. The more upright position of the trunk and subsequently the more upright position of the neck and head may eventually decrease the EMG activity of UT as usually occurs in cases where the head is in a less projected position [41], only to be increased again with a 6-times increase the I-NBV eventually reaching the level of activation shown by females with M-NBV.

Upper trapezius activation could also have been partially increased by the bra worn by the participants, in response to the transfer of breast weight from the pectoral fascia to the UT region via the bra straps [43]. This response has been justified as the need for a stronger muscle contraction to elevate the shoulder against the downward force exerted by the mass of the breast. Other authors reported that wearing a brassiere, particularly when it is tight, may disturb the muscle equilibrium of the pectoral girdle leading to developing myalgia in the area. In a case like this, the contribution of UT in sustaining the breast in a stable elevated position may be increased to counteract the friction developed by the pressure applied by the straps of the brassiere as well as the mass of the breasts [44]. Although the forces developed in the shoulder region were not recorded in the present study, it cannot be ruled out that the shorter projection of the implants used in females with M-NBV, may have ultimately shifted the center of mass of the upper trunk less forward creating a more downward-directed bending moment. The upper trapezius may have eventually responded to this event by generating the force necessary to counteract the added mass through the elastic, non-adjustable bra worn by the participants to keep the external implants in place, increasing its activity [43,44].

A similar pattern of muscle activation was demonstrated between females with I-NBV and M-NBV during TISI45°, but the differences between the groups were not significant. This was probably occurred as the trunk had to be inclined more during TISI45° (45°) compared to SIST (≈30°) [42] thus bringing on some occasions the bulkier breast implants into slight contact with the anterior surface of the thighs, particularly in participants with shorter trunk length. This may partially prevent trunk extensors from generating a higher activity, diminishing therefore the differences between the two groups. Apparently, this was not the case during the execution of SIST as the relative less inclination of the trunk [42] prevented any contact between body parts allowing the thoracic trunk extensors to counterbalance the moment generated by the inclined trunk and the freely suspended breast implants without any restriction.

### 4.1. Clinical Implications

Clinically, the increased trunk muscle responses elicited during everyday movements, as a result of increased breast volume and ultimately breast mass, may account for many of the health problems experienced by a female with enlarged breasts throughout her life. For example, the increased effort of the paraspinal muscles to counteract the increased breast mass, which in our study was manifested by an increase in EMG activity of the TES and LES, up to 28.7% and 18.4%, respectively when I-NBV increase more than 4.5 times can directly affect the blood flow of these muscles. Indeed, blood flow can be restricted when the effort generated by the trunk extensors muscles reaches up to 20% of maximal isometric voluntary contraction (MIVC) and it is thought that if it is applied for a long time, it may cause muscle dysfunction and ultimately upper and lower back pain [45]. Such symptoms are likely to occur in obese females or females who have undergone breast augmentation surgery, as both conditions are associated with an increase in breast volume/mass [7,8,14] leading inevitably in increasing trunk extensor EMG activity.

The increased paraspinal muscle activity obtained in large-breasted females when performing certain daily movements is also expected to cause muscle fatigue if sustained over a long period of time. Even though the paraspinal muscles are highly vascularized [46] and thus better suited to lumbar activities that require high levels of muscular endurance, they can be fatigued by producing only 2% of their maximal voluntary contraction [47]. Muscle fatigue, in addition to the acute local discomfort that it causes, can indirectly affect the passive structures of the posterior trunk by increasing the loads they must withstand. Ultimately the plastic deformation that the posterior capsuloligamentous structures of the trunk suffer over time may lead to the adoption of postures, which in females with large breasts is usually manifested in the upright standing posture with forward trunk inclination [12] and increased thoracic kyphosis [13,14]. These postural changes elicited in females with large breasts are often associated with decreased shoulder elevation range-of-motion [15] and decreased scapular retraction endurance-strength [15] enhancing further the dysfunction and pain symptoms in the neck and upper torso region [15]. They can also cause other postural changes, such as forward head projection and rounded shoulders [48], which in turn can affect the function of other muscles in the area. In general, the forward head projection is thought to increase the EMG activity of the upper neck muscles such as UT [41], and therefore its presence in cervicogenic headache patients cannot be considered accidental [49]. This response, which was noted in the present study in the females with a 6-fold increase in I-NBV through the increase in upper trapezius EMG activity, particularly during SIST could partially explain the head and neck pain symptoms also experienced by females with large breasts [50]. This is not unexpected as the fibers of the upper (descending) part of the trapezius muscle are vulnerable to ischemia due to the significantly lower frequency of type I fibers (58%), at least compared to males (69%), and the inferior capillarization and mitochondrial volume density reported for females [51]. It is again worth noting that females with M-NBV, regardless of how much their breast volume increased, showed similar increased activity of the upper part of the trapezius to that elicited in females with a 6-fold increase in I-NBV. If increased muscle activity accounts to some extent for the cause of shoulder pain, then it should not be considered a coincidence that females with relatively small breast volume experience pain in the area more often than females with larger breast volume [44]. This finding has been attributed to the increased pressure the bra strap puts on the shoulder area, as many females with small breasts tend to accentuate their breast size by wearing tight bras [44]. Our evidence, however, suggests that performing common everyday movements, such as trunk inclination from standing and sitting as well as standing to sit and sit to standing maneuvers can exacerbate upper trapezius EMG activity potentially inducing clinical symptoms in the shoulder region.

Moreover, possible fatigue of the lumbar extensor muscles can impair postural control, which is usually detected via an increase in body sway. This response has been observed in females with breast hypertrophy, who showed increased body sway through the increased area and greater velocity of the center of pressure displacement in the anteroposterior direction under various static and dynamic conditions compared to females without breast hypertrophy [52]. The fact that hypertrophic breasts are associated with reduced postural control was confirmed when postural control in these females was restored after breast reduction surgery [53].

Considering the potential contribution of increased EMG activity of trunk muscles to the induction of painful syndromes and postural adaptations, the need to take measures to minimize or even prevent them before resorting to more radical therapeutic methods (e.g., breast reduction surgery) becomes apparent. Research evidence support that some of the countermeasures that could be taken are increasing the muscular endurance of the extensor muscles of the trunk to withstand the breast mass against gravity [15], regular exercise, and modifications of eating habits to limit body mass and therefore the volume/mass of the breast [14] as well as the selection of appropriate sized bras. In this way, a female may be able to tolerate and manage her oversized breasts during daily and athletic activities.

### 4.2. Study Limitations

Study results should be viewed in the light of some limitations mainly related to the classification of the study sample and the characteristics of the experimental procedure as they may prevent the generalizability of our findings. The study sample was classified based on females having minimal or ideal breast volume [6,26] considering that muscle responses will vary depending on the musculoskeletal adaptations achieved with natural breast volume. Furthermore, the design of the present study required breast volume to be increased proportionally to the natural breast volume of the participants. This decision was based on the fact that the volume of the breast may increase relative to its natural volume as in the case of weight gain [5]. However, our findings may have been different if breasts were artificially augmented regardless of the natural breast volume. Such cases may be faced in plastic surgery when an implant size is chosen based on the doctor’s recommendations and/or the aesthetic satisfaction of the patient [7,8].

The implants also were applied externally and therefore were not expected to affect anatomical structures as may occur with surgically inserted implants or natural breasts. Breast implants inserted surgically create a downward displacement of the breast proportional to the mass of the implant [10] while natural breasts exert tension on the shoulder girdle through the pectoral fascia onto which they are anchored [54]. In both cases, the activation of the neighboring muscles such as the UT may have been increased as a greater activity may be required to counterbalance the exerted forces. Moreover, the results would be different if the long-term, rather than the acute effects of artificially induced breast augmentation, were determined in more, and with greater dispersion in terms of natural breast volume, subjects.

## 5. Conclusions

The evidence of this study showed that performing common daily movements does not elicit the dynamic response of the paraspinal muscles, namely the TES and LES, unless the volume of each breast exceeds approximately 1000 mL. On the other hand, females with smaller breast volumes demonstrated different activation patterns of neck and trunk muscles compared to females with larger breasts in some of the movements tested.

In light of these findings, it also appears that appropriate exercises to increase extensor muscle endurance and limit body weight as well as the selection of appropriate bras could be considered as countermeasures to the adverse effects resulting from increased breast volume. This information may also be useful in breast augmentation surgery, providing new insights into the selection of breast implants in terms of minimizing the risk of developing musculoskeletal problems.

## Figures and Tables

**Figure 1 jfmk-07-00080-f001:**
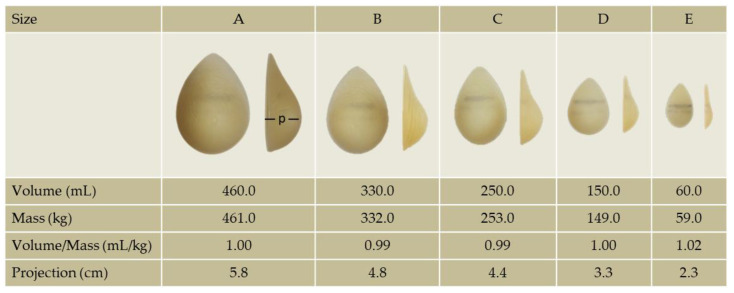
The sizes (**A**–**E**) and corresponding volumes, masses, volume/mass ratios and projections (p) of the external breast implants used in the study.

**Figure 2 jfmk-07-00080-f002:**
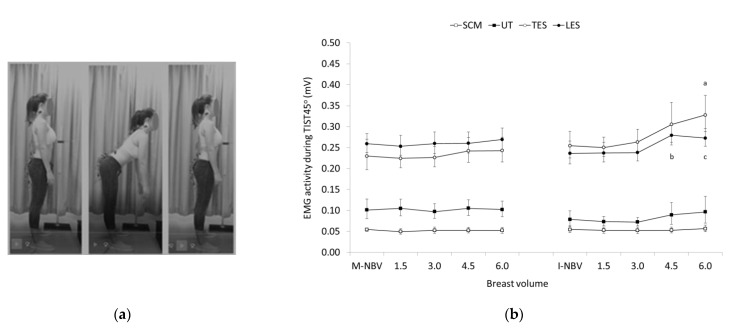
(**a**) The trunk inclination at 45° from the upright standing posture (TIST45°); (**b**) Geometric means and confidence intervals (error bars) of the sternocleidomastoid (SCM), upper trapezius (UT), and thoracic and lumbar erector spinae (TES and LES) EMG activity in females with minimal and ideal natural breast volume (M-NBV, I-NBV) as well as with increased NBV by 1.5-, 3.0-, 4.5- and 6.0-times during TIST45°. ^a^ significant different (SD) compared to I-NBV (p < 0.001), 1.5- (*p* < 0.001) and 3.0-times the I-NBV (*p* < 0.05) for TES; ^b^ SD compared to I-NBV (*p* < 0.01), 1.5- (*p* ≤ 0.05) and 3.0- times the I-NBV (*p* ≤ 0.05) for LES; ^c^ SD compared to I-NBV (*p* < 0.05), 1.5- (*p* ≤ 0.001), and 3.0-times the I-NBV (*p* ≤ 0.05) for LES.

**Figure 3 jfmk-07-00080-f003:**
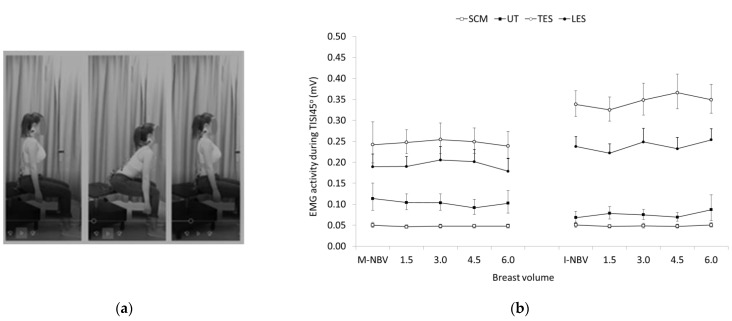
(**a**) The trunk inclination at 45° from the upright sitting posture (TISI45°); (**b**) Geometric means and confidence intervals (error bars) of the sternocleidomastoid (SCM), upper trapezius (UT), and thoracic and lumbar erector spinae (TES and LES) EMG activity in females with minimal and ideal natural breast volume (M-NBV, I-NBV) as well as with increased NBV by 1.5-, 3.0-, 4.5- and 6.0-times during TISI45°.

**Figure 4 jfmk-07-00080-f004:**
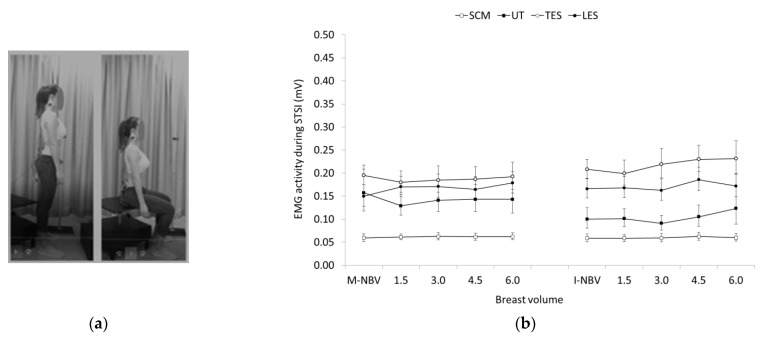
(**a**) Stand-to-Sit (STSI); (**b**) Geometric means and confidence intervals (error bars) of the sternocleidomastoid (SCM), upper trapezius (UT), and thoracic and lumbar erector spinae (TES and LES) EMG activity in females with minimal and ideal natural breast volume (M-NBV, I-NBV) as well as with increased NBV by 1.5-, 3.0-, 4.5- and 6.0-times during STSI.

**Figure 5 jfmk-07-00080-f005:**
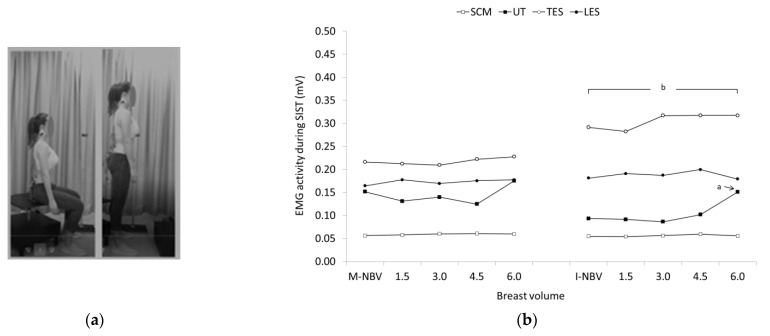
(**a**) Sit-to-Stand (SIST); (**b**) Geometric means and confidence intervals (error bars) of the sternocleidomastoid (SCM), upper trapezius (UT), and thoracic and lumbar erector spinae (TES and LES) EMG activity in females with minimal and ideal natural breast volume (M-NBV, I-NBV) as well as with increased NBV by 1.5-, 3.0-, 4.5- and 6.0-times during sit-to-stand (SIST). ^a^ significant different (SD) compared to females with I-NBV (*p* < 0.05), 1.5- (*p* < 0.05), 3.0- (*p* < 0.01) and 4.5-times (*p* < 0.05) the I-NBV for UT; ^b^ SD compared to females with M-NBV, 1.5-, 4.5- and 6.0-times the M-NBV (*p* ≤ 0.05) for TES.

**Table 1 jfmk-07-00080-t001:** The mean ± standard deviations of demographic and anthropometric characteristics of the females with minimal and ideal natural breast volumes.

Characteristic	M-NBV (*n* = 13)	I-NBV (*n* = 11)
Age (years)	23.92 ± 4.46	24.55 ± 5.68
Height (m)	1.66 ± 0.05	1.63 ± 0.06
Body mass (kg)	58.77 ± 4.44	57.87 ± 5.91
Body Mass Index (kg·m*^−2^*)	21.25 ± 1.58	21.70 ± 1.67
Breast volume (mL)	81.54 ± 34.84	229.55 ± 21.96

M-NBV: minimal natural breast volume; I-NBV: ideal natural breast volume.

**Table 2 jfmk-07-00080-t002:** A description of the movements performed during the experimental protocol.

Movements	Description	
TIST45°		
Starting position	Upright standing posture	with trunk straight, arms hanging freely on the side and eyes facing forward
Transition to final position	Forward inclination of the trunk	
Final position	45° of trunk inclination	
Transition to starting position	Backward inclination of the trunk	
TISI45°		
Starting position	Upright sitting posture	with trunk straight, arms hanging freely on the side and eyes facing forward
Transition to final position	Forward inclination of the trunk	
Final position	45° of trunk inclination	
Transition to starting position	Backward inclination of trunk	
STSI and SIST		
Starting position	Upright standing posture	with trunk straight, arms hanging freely on the side and eyes facing forward
Transition to final position	Lowering to the upright sitting posture	
Starting position	Upright sitting posture	
Transition to starting position	Rising to the upright standing posture	

**Table 3 jfmk-07-00080-t003:** The means ± standard deviations and relative changes (augmented/natural) achieved (in parentheses) in the volume, mass, and projection of the natural and the artificially augmented by 1.5-, 3.0-, 4.5-, and 6.0-times breast, in females with minimal and ideal breast volume.

Group	BIV	TBV	TBM	BIP	TBP
	(mL)	(mL)	(gr)	(cm)	(cm)
M-NBV	-	81.5 ± 34.8 (NBV)	81.5 ± 34.8 (NBM)	-	2.1 ± 0.4 (NBP)
	40.8 ± 17.4	122.3 ± 52.3 (1.5)	122.3 ± 52.3 (1.5)	0.7 ± 1.1	2.8 ± 1.3 (1.3 ± 0.4)
	163.1 ± 69.7	244.6 ± 104.5 (3.0)	244.6 ± 104.5 (3.0)	3.2 ± 0.9	5.3 ± 1.3 (2.5 ± 0.3)
	285.4 ± 122.0	366.9 ± 156.8 (4.5)	366.9 ± 156.8 (4.5)	4.4 ± 1.0	6.5 ± 1.3 (3.0 ± 0.3)
	407.7 ± 174.2	489.2 ± 209.1 (6.0)	489.2 ± 209.1 (6.0)	5.5 ± 1.4	7.7 ± 1.7 (3.6 ± 0.5)
I-NBV	-	229.5 ± 22.0 (NBV)	229.5 ± 22.0 (NBM)	-	3.3 ± 0.3 (NBP)
	114.8 ± 11.0	344.3 ± 32.9 (1.5)	344.3 ± 32.9 (1.5)	2.2 ± 0.0	5.5 ± 0.3 (1.7 ± 0.1)
	459.1 ± 43.9	688.6 ± 65.9 (3.0)	688.6 ± 65.9 (3.0)	5.8 ± 0.1	9.1 ± 0.3 (2.8 ± 0.1)
	803.4 ± 76.9	1033.0 ± 98.8 (4.5)	1033.0 ± 98.8 (4.5)	8.2 ± 0.8	11.5 ± 0.1 (3.5 ± 0.2)
	1147.7 ± 109.8	1377.3 ± 131.8 (6.0)	1377.3 ± 131.8 (6.0)	11.5 ± 1.2	14.7 ± 1.4 (4.5 ± 0.3)

M-NBV = Minimal natural breast volume; I-NBV = Ideal natural breast volume; BIV = Breast implant volume; TBV = Total breast volume; TBM = Total breast mass; BIP = Brest implant projection; TBP = Total breast projection; NBV = Natural breast volume; NBM = Natural breast mass; NBP = Natural breast projection.

## Data Availability

The data presented in this study are available on request from the corresponding author.

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
