# Peer review of "Effects of Artificially Induced Breast Augmentation on the Electromyographic Activity of Neck and Trunk Muscles during Common Daily Movements"

_jfmk, 2022, doi:10.3390/jfmk7040080_

Round 1
Reviewer 1 Report
The authors present an interesting and very original paper investigating the effects of an artificial increase of breasts size on EMG activities of neck and trunk muscles during specific movements.
This is an area that has received a very little attention in the literature, therefore, warrants further examination. Overall, the manuscript is written and organized fairly well. It follows the logical sequence of a research purpose. Despite this strength, I have some comments that need to be addressed by the authors and listed below.
INTRODUCTION
First and second paragraph should be merged and synthetized.
MATERIAL and METHODS
Line 81 : “out of 58 females invited” brings nothing and should be suppressed.
Line 82: Minimal and ideal natural breast volumes should be further defined to understand the difference between both groups and inclusion criteria related to the volume of the breasts.
Table1. Body height should be expressed with two decimal. The last column of the table entitled "Total" is useless.
Figure 1. Units for Volume and Mass are missing.
Line 112 : Authors should precise if trials were performed randomly or by block of condition related to the increase of breast volumes.
RESULTS
The authors present their results observing differences between groups with mixed ANOVA.
In parallel, authors should explore if there is a relationship between initial breast volumes, the different increase of breast volumes and EMG activities for the concerned muscles for all subjects.
DISCUSSION
The discussion is very interesting. However, authors should extend their results with postural control, upright standing, excess of body weight and induced fatigue models.
From clinical/preventive perspectives, their results should be more linked with potential musculoskeletal disorders and women with large natural or artificial breasts.
REFERENCES
The number of references (n=58) is very high for a research article. Some references are useless and/or redundant.
Author Response
Dear Editor-in-Chief, Dear Reviewer
Thank you for giving me the opportunity to review the manuscript entitled "Effects of Artificially Induced Breast Augmentation on the Electromyographic Activity of Neck and Trunk Muscles during Common Daily Movements”. All comments and questions that were answered and presented below are highlighted in yellow. Sections highlighted in gray represent responses addressing comments made by both reviewers.
Responses to 1st reviewer
The authors present an interesting and very original paper investigating the effects of an artificial increase of breast size on EMG activities of neck and trunk muscles during specific movements. This is an area that has received a very little attention in the literature, therefore, warrants further examination. Overall, the manuscript is written and organized fairly well. It follows the logical sequence of a research purpose. Despite this strength, I have some comments that need to be addressed by the authors and listed below.
Comment
INTRODUCTION: First and second paragraph should be merged and synthetized.
Response
The first and second paragraph were merged and synthetized as follows:
Lines 41-48: “…However, although breast volume is of most concern to females as it relates to their image and aesthetics, it is the weight of the breast that can affect their skeletal and soft tissue structures, especially in females with large breasts, causing musculoskeletal pain syndromes and dysfunction. Considering that the breast’s mass-to-volume ratio is close to one for both premenopausal (1.07 g/mL) and postmenopausal breasts (1.06 g/ml) [9] it becomes apparent that the load added by the breasts on the female's chest may vary from a few grams to several kilograms depending on their volume.”
Comment
MATERIAL and METHODS: Line 81: “out of 58 females invited” brings nothing and should be suppressed.
Response
Line 84: The phrase “out of 58 females invited” was deleted as requested by the reviewer.
Comment
Line 82: Minimal and ideal natural breast volumes should be further defined to understand the difference between both groups and inclusion criteria related to the volume of the breasts.
Response
Breast volumes were further defined as suggested by the reviewer:
Lines 90-97: “The M-NBV group included females whose volume of each breast ranged between 50 mL and 150 mL. It was assumed that the breast volume in this group approximated the small breast volume experienced by females with micromastia or after mastectomy. Females with I-NBV were considered those with a breast volume between 200 mL and 250 mL. The breast volume of these subjects was within the limits of what is considered ideal (200 mL - 350 mL) based on clinical observations of the volumes used to fill the implants in breast reconstruction surgery [6,23].”.
Comment
Table 1. Body height should be expressed with two decimals. The last column of the table entitled "Total" is useless.
Response
Table 1: Body height was expressed with two decimals as suggested by the reviewer. For consistency, all variables presented in this table are expressed with two decimals.
Comment
Figure 1. Units for Volume and Mass are missing.
Response
Figure 1: The units for volume and mass were added as requested by the reviewer
Comment
Line 112: Authors should precise if trials were performed randomly or by block of condition related to the increase of breast volumes.
Response
The randomization technique used in the present study was added as follows:
Lines 129-133: “The order in which the movements were performed as well as the order in which the breast volume increased to perform these movements was randomly determined by generating two sets of random sequences using an online application (https://www.random.org accessed on 20 December 2021).”
Comment
RESULTS: The authors present their results observing differences between groups with mixed ANOVA. In parallel, authors should explore if there is a relationship between initial breast volumes, the different increase of breast volumes and EMG activities for the concerned muscles for all subjects.
Response
Pearson correlation coefficients between normal and increased breast volume and EMG activities recorded during TIST45, TISI45, STSI and SIST for the SCM, UT, TTE and LTE muscles were calculated according to the reviewer's suggestion. Calculations revealed no significant correlations between NBVs (as well as increased breast volumes, presumably because they were multiples of the NBV, and EMG activities for the muscles examined in all movements (r = -0.438 - 0.278). With the proposed investigation it was possible to determine the relationships between the two variables (breast volume/weight and EMG activity), the existence of which would indicate that the activation of the specific muscles depended to some extent on the breast mass. The lack of statistical correlation between them indicated that such a relationship cannot be proven from the data of the present study. A possible reason for this may be the small range of dispersion of breast volume values ​​as this limitation would hide a potential relationship between the two factors, even if a true relationship existed. Given this limitation, the authors of this study would prefer not to present this exploratory approach to the data. However, we would like to thank the reviewer for bringing this statistical approach to our attention as it will be considered in the studies that are already being done in our laboratory and are expected to be published in the near future.
Comment
- DISCUSSION: The discussion is very interesting. However, authors should extend their results with postural control, upright standing, excess of body weight and induced fatigue models.
- From clinical/preventive perspectives, their results should be more linked with potential musculoskeletal disorders and women with large natural or artificial breasts.
Response
Lines 367-425: A separate paragraph under the subtitle “4.1. Clinical implications” was added discussing the results of the study and the potential clinical outcomes associated with them. The paragraph was highlighted in grey color as it contains information that is suggested by both reviewers.
“Clinically, the increased trunk muscle responses elicited during everyday movements, as a result of increased breast volume and ultimately breast mass, may account for many of the health problems experienced by a female with enlarged breasts throughout her life. For example, the increased effort of the paraspinal muscles to counteract the increased breast mass, which in our study was manifested by an increase in EMG activity of the TES and LES, up to 28.7% and 18.4%, respectively when I-NBV increase more than 4.5 times can directly affect the blood flow of these muscles. Indeed, blood flow can be restricted when the effort generated by the trunk extensors muscles reaches up to 20% of maximal isometric voluntary contraction (MIVC) and it is thought that if it is applied for a long time, it may cause muscle dysfunction and ultimately upper and lower back pain [45]. Such symptoms are likely to occur in obese females or females who have undergone breast augmentation surgery, as both conditions are associated with an increase in breast volume/mass [7,8,14] leading inevitably in increasing trunk extensor EMG activity.
The increased paraspinal muscle activity obtained in large-breasted females when performing certain daily movements is also expected to cause muscle fatigue if sustained over a long period of time. Even though the paraspinal muscles are highly vascularized [46] and thus better suited to lumbar activities that require high levels of muscular endurance, they can be fatigued by producing only 2% of their maximal voluntary contraction [47]. Muscle fatigue, in addition to the acute local discomfort that it causes, can indirectly affect the passive structures of the posterior trunk by increasing the loads they must withstand. Ultimately the plastic deformation that the posterior capsuloligamentous structures of the trunk suffer over time may lead to the adoption of postures, which in females with large breasts is usually manifested in the upright standing posture with forward trunk inclination [12] and increased thoracic kyphosis [13,14]. These postural changes elicited in females with large breasts are often associated with decreased shoulder elevation range-of-motion [15], and decreased scapular retraction endurance-strength [15] enhancing further the dysfunction and pain symptoms in the neck and upper torso region [15]. They can also cause other postural changes, such as forward head projection and rounded shoulders [48], which in turn can affect the function of other muscles in the area. In general, the forward head projection is thought to increase the EMG activity of the upper neck muscles such as UT [49], and therefore its presence in cervicogenic headache patients cannot be considered accidental [50]. This response, which was noted in the present study in the females with a 6-fold increase in I-NBV through the increase in upper trapezius EMG activity, particularly during SIST could partially explain the head and neck pain symptoms also experienced by females with large breasts [51]. This is not unexpected as the fibers of the upper (descending) part of the trapezius muscle are vulnerable to ischemia due to the significantly lower frequency of type I fibers (58%), at least compared to males (69%), and the inferior capillarization and mitochondrial volume density reported for females [52]. It is again worth noting that females with M-NBV, regardless of how much their breast volume increased, showed similar increased activity of the upper part of the trapezius to that elicited in females with a 6-fold increase in I-NBV. If increased muscle activity accounts to some extent for the cause of shoulder pain, then it should not be considered a coincidence that females with relatively small breast volume experience pain in the area more often than females with larger breast volume [44]. This finding has been attributed to the increased pressure the bra strap puts on the shoulder area, as many females with small breasts tend to accentuate their breast size by wearing tight bras [44]. Our evidence, however, suggests that performing common everyday movements, such as trunk bending from standing and sitting as well as standing to sit to standing maneuvers can exacerbate upper trapezius EMG activity potentially inducing clinical symptoms in the shoulder region.
Moreover, possible fatigue of the lumbar extensor muscles can impair postural control, which is usually detected via an increase in body sway. This response has been observed in females with breast hypertrophy, who showed increased body sway through the increased area and greater velocity of the center of pressure displacement in the anteroposterior direction under various static and dynamic conditions compared to females without breast hypertrophy [53]. The fact that hypertrophic breasts are associated with reduced postural control was finally confirmed when postural control in these females was restored after breast reduction surgery [54].”
Comment
REFERENCES: The number of references (n=58) is very high for a research article. Some references are useless and/or redundant.
Response: A considerable effort was made to reduce the number of references used in the present study. Up to a point this was achieved by reducing the number of references from 58 to 46. However, the information that was added to support the comments proposed by the reviewer forces us to include 9 more references, reaching a total number of 55. The references that were used for the Introduction were 25, 8 were used to support the methodology, and 22 to justify the information that was required to be discussed in the Discussion section. The authors agree that the number of references is large, however, they feel obliged to refer to the names of all those who have supplied the necessary inform

Reviewer 2 Report
This manuscript present research regarding the effect of artificially induced breast volume on muscular activity of neck and trunk musculature during everyday movements. Indeed, there is lack of research in this field which makes this manuscript interesting to the readers. These are my comments and suggestions:
Abstract:
Abstract is nicely written. My only comment is: "during stand-to-site" - it should be "stand-to-sit".
Introduction:
Introduction is adequate, with clearly stated purpose of the study. Theoretical background includes all the relevant information supported by references.
Methods:
Please state which university - e.g. "University’s Human Research Ethics Committee". How did you estimate your sample size? Line 97: "undergo" - I suggest to use past tense.
Results:
No comments.
Discussion and conclusion:
I suggest to expand the text a bit regarding the possible difference between acute changes of EMG activity (your research), and possible adaptation to larger breasts during longer time. Also, add the paragraph regarding possible countermeasures to facilitate adaptation (in terms of prevention of pain symptoms or postural problems), add some clinical perspective.
Author Response
Dear Editor-in-Chief, Dear Reviewer
Thank you for giving me the opportunity to review the manuscript entitled "Effects of Artificially Induced Breast Augmentation on the Electromyographic Activity of Neck and Trunk Muscles during Common Daily Movements”. All comments and questions that were answered and presented below are highlighted in cyan. Sections highlighted in gray represent responses addressing comments made by both reviewers.
Responses to 2nd reviewer
This manuscript present research regarding the effect of artificially induced breast volume on muscular activity of neck and trunk musculature during everyday movements. Indeed, there is lack of research in this field which makes this manuscript interesting to the readers. These are my comments and suggestions:
Comment
Abstract: Abstract is nicely written. My only comment is: "during stand-to-site" - it should be "stand-to-sit".
Response
Line 15: Change was made as suggested by the reviewer
Comment
Introduction: Introduction is adequate, with clearly stated purpose of the study. Theoretical background includes all the relevant information supported by references.
Response: N/A
Comment
Methods: Please state which university - e.g., "University’s Human Research Ethics Committee". How did you estimate your sample size? Line 97: "undergo" - I suggest to use past tense.
Response
Changes were made as suggested by the reviewer
Lines 86-90: “The sample size was determined based on an a priori power analysis aiming to achieve a statistical power of 80%, with an effect size (f) = 0.25 (calculated based on a partial η2 = 0.06) and a statistical significance of a = 0.05, by implementing a free accessed statistical application (G*Power v. 3.1.9.2; FranzFaul, Universität Kiel, Germany, accessed on 5 January 2022).”
Lines 101-102: “…National and Kapodistrian University of Athens’s…”
Line 111: “..underwent…” instead “…undergo…”
Comment
Results: No comments.
Response: N/A
Comment
Discussion and conclusion: I suggest to expand the text a bit regarding the possible difference between acute changes of EMG activity (your research), and possible adaptation to larger breasts during longer time. Also, add the paragraph regarding possible countermeasures to facilitate adaptation (in terms of prevention of pain symptoms or postural problems), add some clinical perspective.
Response:
Lines 367-425: A separate paragraph under the subtitle “4.1. Clinical implications” was added discussing the results of the study and the potential clinical outcomes associated with them. The paragraph was highlighted in grey color as it contains information that is suggested by both reviewers.
“Clinically, the increased trunk muscle responses elicited during everyday movements, as a result of increased breast volume and ultimately breast mass, may account for many of the health problems experienced by a female with enlarged breasts throughout her life. For example, the increased effort of the paraspinal muscles to counteract the increased breast mass, which in our study was manifested by an increase in EMG activity of the TES and LES, up to 28.7% and 18.4%, respectively when I-NBV increase more than 4.5 times can directly affect the blood flow of these muscles. Indeed, blood flow can be restricted when the effort generated by the trunk extensors muscles reaches up to 20% of maximal isometric voluntary contraction (MIVC) and it is thought that if it is applied for a long time, it may cause muscle dysfunction and ultimately upper and lower back pain [45]. Such symptoms are likely to occur in obese females or females who have undergone breast augmentation surgery, as both conditions are associated with an increase in breast volume/mass [7,8,14] leading inevitably in increasing trunk extensor EMG activity.
The increased paraspinal muscle activity obtained in large-breasted females when performing certain daily movements is also expected to cause muscle fatigue if sustained over a long period of time. Even though the paraspinal muscles are highly vascularized [46] and thus better suited to lumbar activities that require high levels of muscular endurance, they can be fatigued by producing only 2% of their maximal voluntary contraction [47]. Muscle fatigue, in addition to the acute local discomfort that it causes, can indirectly affect the passive structures of the posterior trunk by increasing the loads they must withstand. Ultimately the plastic deformation that the posterior capsuloligamentous structures of the trunk suffer over time may lead to the adoption of postures, which in females with large breasts is usually manifested in the upright standing posture with forward trunk inclination [12] and increased thoracic kyphosis [13,14]. These postural changes elicited in females with large breasts are often associated with decreased shoulder elevation range-of-motion [15], and decreased scapular retraction endurance-strength [15] enhancing further the dysfunction and pain symptoms in the neck and upper torso region [15]. They can also cause other postural changes, such as forward head projection and rounded shoulders [48], which in turn can affect the function of other muscles in the area. In general, the forward head projection is thought to increase the EMG activity of the upper neck muscles such as UT [49], and therefore its presence in cervicogenic headache patients cannot be considered accidental [50]. This response, which was noted in the present study in the females with a 6-fold increase in I-NBV through the increase in upper trapezius EMG activity, particularly during SIST could partially explain the head and neck pain symptoms also experienced by females with large breasts [51]. This is not unexpected as the fibers of the upper (descending) part of the trapezius muscle are vulnerable to ischemia due to the significantly lower frequency of type I fibers (58%), at least compared to males (69%), and the inferior capillarization and mitochondrial volume density reported for females [52]. It is again worth noting that females with M-NBV, regardless of how much their breast volume increased, showed similar increased activity of the upper part of the trapezius to that elicited in females with a 6-fold increase in I-NBV. If increased muscle activity accounts to some extent for the cause of shoulder pain, then it should not be considered a coincidence that females with relatively small breast volume experience pain in the area more often than females with larger breast volume [44]. This finding has been attributed to the increased pressure the bra strap puts on the shoulder area, as many females with small breasts tend to accentuate their breast size by wearing tight bras [44]. Our evidence, however, suggests that performing common everyday movements, such as trunk bending from standing and sitting as standing to sit and sit to standing maneuvers can exacerbate upper trapezius EMG activity potentially inducing clinical symptoms in the shoulder region.
Moreover, possible fatigue of the lumbar extensor muscles can impair postural control, which is usually detected via an increase in body sway. This response has been observed in females with breast hypertrophy, who showed increased body sway through the increased area and greater velocity of the center of pressure displacement in the anteroposterior direction under various static and dynamic conditions compared to females without breast hypertrophy [53]. The fact that hypertrophic breasts are associated with reduced postural control was finally confirmed when postural control in these females was restored after breast reduction surgery [54].”
Lines 426-435: A paragraph regarding possible countermeasures to facilitate adaptation (in terms of prevention of pain symptoms or postural problems) was also added as suggested by the reviewer.
“Considering the potential contribution of increased EMG activity of trunk muscles to the induction of painful syndromes and postural adaptations, the need to take measures to minimize or even prevent them before resorting to more radical therapeutic methods (e.g., breast reduction surgery) becomes apparent. Research evidence support that some of the countermeasures that could be taken are increasing the muscular endurance of the extensor muscles of the trunk to withstand the breast mass against gravity [15], regular exercise, and modifications of eating habits to limit body mass and therefore the volume/mass of the breast [14] as well as the selection of appropriate sized bras. In this way, a female may be able to tolerate and manage her oversized breasts during daily and athletic activities.”
Lines 465-470: Conclusions were modified to comply with the reviewer’s comments
“In light of these findings, it also appears that appropriate exercises to increase extensor muscle endurance and to limit body weight as well as the selection of appropriate bras could be considered as countermeasures to the adverse effects resulting from increased breast volume.”

Round 2
Reviewer 1 Report
All my comments have been considered by the authors.
This new version of the manuscript has been improved compared to its initial version.